# Mid-Infrared Broadband Achromatic Metalens with Wide Field of View

**DOI:** 10.3390/ma15217587

**Published:** 2022-10-28

**Authors:** Yurong Jiang, Cheng Cui, Jinmin Zhao, Bin Hu

**Affiliations:** Beijing Engineering Research Center for Mixed Reality and Advanced Display, School of Optics and Photonics, Beijing Institute of Technology, Beijing 100081, China

**Keywords:** metalens, monochromatic aberration-corrected, achromatic

## Abstract

Metasurfaces have the ability to flexibly control the light wavefront, and they are expected to fill the gaps of traditional optics. However, various aberrations pose challenges for the application of metasurfaces in the wide angle and wide spectral ranges. The previous multi-aberration simultaneous optimization works had shortcomings such as large computational load, complex structure, and low generality. Here, we propose a metalens design method that corrects both monochromatic and chromatic aberrations simultaneously. The monochromatic aberration-corrected phase distribution is obtained by the optical design, and the chromatic aberration is reduced by using the original search algorithm combined with dispersion engineering. The designed single-layered wide-angle achromatic metalens has a balanced and efficient focusing effect in the mid-infrared band from 3.7 μm to 5 μm and a wide angle of ±30°. The design method proposed has the advantages of low computational load, wide application range, and easy experimental fabrication, which provides new inspiration for the development of generalized software for the design and optimization of metasurfaces.

## 1. Introduction

Metasurfaces have attracted the attention of researchers all over the world due to the growing demand for light, miniaturized, and highly integrated optical devices. Metasurfaces can arbitrarily modulate the amplitude, phase, and polarization state [1,2,3,4] of the optical field with subwavelength spatial resolution through well-arranged nanoantenna elements. Compared with traditional refractive optical elements, metasurfaces have the advantages of ultralight weight and high compactness [5]. As an important development direction of future optical devices, the design and fabrication capabilities of metasurfaces have been rapidly improved in the past decade, and various metasurface-based functional devices such as lenses [6,7], vortex beam generators [8,9], polarization-resolving devices [10], retroreflectors [11], orbital angular momentum (OAM) superposition-state generators [12], and meta-waveplates [13] have been proposed.

However, chromatic aberration is still a common problem for metalenses. The main reason for the existence of chromatic aberration of the metalens is the quasi-periodic structure caused by 2π phase wrapping [14]. The metalens behaves as a grating with radially varying period. According to the grating equation, the metalens deflects different incident wavelengths by different angles. Some scholars have focused on enhancing the dispersion of diffractive metalens. Such a hyper-dispersive metasurface had a larger zoom range than an ordinary metalens and was used to increase the imaging depth of fast spectral tomography [15]. More researchers corrected chromatic aberration by designing metalenses to have different local periods at different wavelengths. Approaches that have been proposed include dispersion compensation for unit cells through dispersion engineering [16], the use of multilayer metasurfaces [17], and full-aperture optimization methods [18]. Among them, dispersion engineering is the most commonly used method to reduce chromatic aberration. This method enables arbitrary control of the metasurface dispersion over a continuous bandwidth by independently constraining the group delay (GD) and the group delay dispersion (GDD) of each unit cell at a specified frequency [19]. To date, several constructive works on achromatic metasurfaces have been reported, covering many different wavebands. Capasso et al. proposed many pioneering design ideas for achromatic metasurfaces in the visible band and reported a series of influential research results. In 2017, they experimentally reported an achromatic metalens working in the reflection mode [20]. The metalens used a single titanium dioxide nanoantenna as a unit cell to achieve achromatic focusing in the 490–550 nm band. One year later, they went a step further and coupled two titania nanofins of different sizes as a unit cell, greatly increasing the degree of freedom in the design of achromatic metasurfaces [19]. The designed metalens was able to achieve achromatic focusing and imaging in the 470–670 nm band with an efficiency of about 20%. In 2019, they reported a polarization-insensitive achromatic metasurface [21]. The metalens employed three nanofins of different sizes coupled to form a unit cell, which enabled diffraction-limited focusing in the 460–700 nm band. Moreover, the focusing efficiency was changed by only 4% under different incident polarizations. For longer optical wavelengths, in 2018, Shrestha et al. reported an achromatic focusing metalens in the near-infrared band [22]. The unit cell of the metasurface contained a variety of silicon nanorods with different shapes and different duty cycles. The metalens achieved achromatic focusing in the 1200–1650 nm band, and the focusing efficiency was up to 50%. Two years later, Ou et al. designed two polarization-controlled multifunctional achromatic metalenses based on elliptical silicon nanorod unit cells in the mid-infrared band [23]. In addition to achieving achromatic focusing in the 3.5–5 μm band, according to the polarization direction of the incident light, the metalens also generated vortex beams with different topological charges or deflected the focus to different directions. In the same year, Zhao et al. proposed an achromatic focusing metalens based on unit cells composed of two silicon nanofins in the terahertz band [24]. The metalens achieved achromatic focusing in the 111–131 μm band, with a focusing efficiency of 13.01–32.6%. These results show the great potential of achromatic metasurfaces in the field of optical imaging in the future.

In addition to the chromatic aberration problem, the metasurface also suffers from severe monochromatic aberration. Although the spherical aberration of metalenses can be eliminated by controlling the phase profile of metasurfaces, other monochromatic aberrations such as coma and astigmatism still hinder the application of metasurfaces in the environment of wide field of view (FOV) [25]. The use of cascaded metasurfaces is the most common method to increase the usable FOV of metalens [25,26,27], but leads to a rather complicated fabrication process. There are also reports that arrays of nanorods are arranged on spherical or flexible substrates to completely eliminate the coma aberration of the system [28,29]. However, these methods need difficult fabrication processes, and the metasurfaces are not flat.

In 2021, there were several reports of metasurfaces for focusing and imaging that considered both a large FOV and dispersion compensation. Sawant et al. proposed a metalens that can simultaneously compensate for dispersion and correct for monochromatic aberrations [30]. The metasurface was closely attached to the exit surface of a plano-convex lens, which reduced the chromatic aberration by 80% and the spherical aberration by 70% of the plano-convex lens. However, because of this, the metasurface cannot be used alone and has centimeter-scale dimensions. This is inconsistent with the development trend of miniaturized optical systems. Yang et al. designed an achromatic and wide-field metalens using free-form meta-atoms generated by deep learning algorithms [31]. The metalens achieved an average focusing efficiency of 45% within a bandwidth of 1–1.2 μm and an FOV of 180°. Unfortunately, the complex configuration of the meta-atom results in a very high computational load, requiring the use of commercial cloud computing services, and it is also very difficult to manufacture experimentally. On the other hand, reports tend to focus on the demonstration of focusing performance of metalens at large FOV rather than systematic optimization. Tseng et al. fabricated a full-color (400–700 nm) and wide-FOV (up to 40°) metasurface imaging system using a convolutional neural network combined with an image reconstruction algorithm [32]. The system had image quality comparable to that of a six-element compound refractive optical system, but was 550,000 times smaller. Despite the excellent imaging quality, the metalens required extensive time for a priori learning in the design process. The metalens was not reconfigurable; thus, it only reproduced the images in the learning set, which further limits its application in more general scenarios.

Although many works for aberration correction of metalenses have been reported, most of these studies focused only on one aberration correction. In this paper, we propose a metalens corrected for both monochromatic and chromatic aberrations. The working band of the metalens is from 3.7 μm to 5 μm, which is located in the mid-infrared band. Mid-infrared spectroscopy has important applications in many fields, such as gas sensing, astrophysics, free-space communication, and molecular fingerprint detection [33]. Traditional mid-infrared optical components are large in size; thus, metasurfaces have broad application prospects on the road of miniaturization and compactness of mid-infrared optical devices. In the design process, we use optical design methods to optimize the monochromatic aberration of the metalens by adopting the phase distribution of the aspherical lens. We also employ an improved direct search algorithm combined with dispersion engineering to correct the chromatic aberration of the metalens. The entire design process is fast and general, which does not require an a priori learning process or commercial computing system. The designed metalens has stable and balanced focusing performance in the entire working band with a large FOV of 60°, and the focusing efficiency is 6.7–42.4%. In addition, it is demonstrated that our metalens has a significant performance advantage over the conventional metalens with only chromatic aberration corrections at a large incident angle of 30°. The proposed method provides a general design idea for metalenses with simultaneous chromatic and monochromatic aberration correction in the mid-infrared or other wavelengths. Due to the low computational load and high generality of our design process, it is expected to drive the emergence of systematic metasurface design and optimization software.

## 2. Principle and Design Method

### 2.1. Principle of Correcting Monochromatic Aberration of Metalens

In the design of traditional metalenses, Equation (1) is usually used to express a monochromatic focusing metalens without spherical aberration [34].
(1)$$\varphi \left(r\right)=-\frac{2\pi }{\lambda }\left(\sqrt{r^{2}+f^{2}}-f\right),$$
where *λ*, *f*, and *r* are the wavelength, the designed focal length, and radial coordinate of the lens, respectively. We used Zemax Optic Studio 2005, a commercial optical design software, to simulate the beam focusing effect of this type of lenses at different incident angles using the ray-tracing technique, and the results are shown in Figure 1a. In the simulation, the incident wavelength λ was set to 4.25 μm, the radius of the lens was 50.5 μm, and the designed focal length f was 200 μm. We can find from Figure 1a that, when the light was incident perpendicular to the lens, the emergent beam could be well focused to a point on the focal plane. However, as the angle θ between the incident beam and the optical axis increased, the focal length of the emergent beam changed, and the focusing effect of the beam on the focal plane continued to deteriorate. As mentioned above, the reason for this phenomenon is that monochromatic aberrations such as coma and astigmatism are still present in the system. Combining Equation (1) with Kirchhoff’s scalar diffraction theory [35], we can quantitatively calculate the intensity distribution of the emergent beam on the focal plane of this type of lens, and we can more intuitively characterize the focusing effect of the beam. Kirchhoff’s scalar diffraction theory can be expressed as
(2)$$E\left(x,y,z\right)=\frac{1}{j\lambda }\iint E\left(x_{0},y_{0},z_{0}\right)\exp\left(jk_{0}l\right)/ldx_{0}dy_{0},$$
where *E* is the electric field component of the emergent linearly polarized light, *j* is the imaginary unit, *x*_0_, *y*_0_, and *z*_0_ are the coordinates on the exit surface of the lens, *k*_0_ is the vacuum wave vector, and $l=\sqrt{{\left(x-x_{0}\right)}^{2}+{\left(y-y_{0}\right)}^{2}+{\left(z-z_{0}\right)}^{2}}$ is the distance between the observation point and the point on the exit surface. We calculated the electric field intensity distribution on the focal plane of the lens when the angle of incidence θ was 0°, 10°, 20°, and 30°, respectively. The results are shown in Figure 1b. The electric field intensity distributions in all pictures are normalized, and the focal points are placed in the central area of the pictures. The actual coordinates of the focal point at each incident angle θ were (0, 0), (35.27 μm, 0), (72.79 μm, 0), and (115.47 μm, 0), respectively. It can be seen from the calculated results that, with the increase in incident angle, the focal point gradually deformed, and the intensity of the focal point decreased continuously. Clearly, the existence of monochromatic aberrations hinders the use of metalens in imaging applications that require a large FOV.

In geometric optics, the most common method of reducing monochromatic aberrations is to use an aspherical lens [36]. On the basis of this method, we used Zemax to design and optimize a lens with two tenth-order even aspheric surfaces, extract its geometric profile, and convert it to the phase distribution of the metalens. This scheme of correcting the monochromatic aberration of the metalens by optical design was verified in our previous work designing a large-angle zoom metalens [37]. The wavelength of the incident light was still 4.25 μm, and the radius of the lens remained at 50.5 μm. The material of the lens was ZnSe, and the refractive index at the designed wavelength was n1 = 2.417. In Zemax, the geometric profile equation of a tenth-order even aspheric surface was computed as
(3)$$h_{i}\left(r\right)=\frac{c_{i}r^{2}}{1+\sqrt{1-\left(1+K_{i}\right)c_{i}^{2}r^{2}}}+\overset{5}{\underset{m=1}{\sum }}a_{m}r^{2m},$$
where *r* is the radial coordinate of the lens, and c, k, and m are the optimizable parameters. The subscripts *i* = 1, 2 represent the two surfaces of the lens. The optimization goal is to minimize the root mean square (RMS) of the focal spot when the incident angle θ reaches ±30°.

The aspheric lens after the design and parameter optimization is shown in Figure 1c. It can be clearly observed from the figure that, when the incident angle θ increased from 0° to 30°, the emergent light maintained a good focusing effect on the focal plane. The optimized parameter values are listed in Table 1. The actual focal length of the optimized lens was 200.014 μm. The thickness at the center of the lens was L = 20.09 μm. It can be seen from Figure 1c that the thickness distribution of the lens can be calculated as follows:(4)$$h_{l}\left(r\right)=L+h_{2}\left(r\right)-h_{1}\left(r\right).$$

The total thickness of the system is given by
(5)$$H=L+\max\left[h_{2}\left(r\right)\right].$$

Therefore, the phase distribution of the monochromatic aberration-corrected metalens can be approximately expressed as follows [37]:(6)$$\varphi \left(r\right)=\frac{2\pi }{\lambda }\left\{n_{0}\left[H-h_{l}\left(r\right)\right]+n_{1}h_{l}\left(r\right)\right\},$$
where $n_{0}=1$ is the refractive index of air. We combined Equation (6) with Kirchhoff’s scalar diffraction theory to calculate the electric field intensity distribution on the focal plane of the lens at different incident angles after reducing monochromatic aberrations, and the results are shown in Figure 1d. Before and after the correction of monochromatic aberration, the actual coordinates of the focal point of the lens at different incident angles θ remained unchanged. As with the processing of Figure 1b, the electric field intensity distributions in all pictures of Figure 1d are normalized, and the focal points are all located in the central area of the pictures. Comparing Figure 1d with Figure 1b, we can clearly see that, at a large FOV, the aberration-optimized lens has the advantages of smaller focal deformation and higher focal intensity than the traditional lens, and the focusing effect under different FOVs is relatively balanced. Therefore, the optimized lens can effectively focus beams over a large FOV of ±30°.

It is worth noting that, in order to make the calculated results of Equation (6) accurate, the transmission of the beam inside the lens under different incident angles should be as parallel to the optical axis as possible, such as the trajectory in Figure 1c. If the transmission of light inside the lens is oblique to the optical axis, the actual propagation distance of the light inside the lens cannot be regarded as the thickness of the lens at that position, resulting in errors in the optical path and phase calculations. Therefore, we carefully control the curvature of the object surface of the lens such that beams at different angles of incidence can travel parallel to the optical axis within the lens. In fact, if the curvature of the object-side surface of the lens is adjusted to be negative, it can be optimized to obtain a smaller RMS of the lens focal spot. However, the beam propagates obliquely in the lens in this situation, resulting in a poor focusing effect of the lens. This problem was better solved in some articles [25,26], i.e., using a metalens doublet. In this scheme, the two nanorod arrays located on the upper and lower sides of the same substrate correspond to the two surfaces of the aspheric lens, respectively. Therefore, it is only necessary to extract the geometric contours of the two surfaces of the lens and convert them into phase distributions. The difficulty is that the emergent surface needs a nanorod array with a larger radius than the incident surface, which is used to completely receive the light passing through the substrate under oblique incidence to avoid the decline of the focusing effect. The difference in the radius of the two nanorod arrays and the alignment between the two layers make the actual fabrication of the metalenses significantly more difficult. Our method employs a single-layered metalens structure based on Equation (6), which results in lower device design and fabrication difficulty, better compactness, and integration, while maintaining high focusing efficiency.

### 2.2. Principle of Dispersion Compensation of Metalens

In order to realize the achromatic focusing function of the metalens, it is not enough to rely only on the single-wavelength phase distribution defined by Equation (6). Dispersion compensation must be performed on the metalens so that the incident light with different wavelengths can be focused on the same focal point. The dispersion of a metalens depends on its spatial- and frequency-dependent phase profile φ (r, ω), where r is the radial coordinate of the metalens, and ω is the angular frequency of the incident light. The phase profile can be Taylor-expanded near the designed angular frequency ω0, expressed as follows [19]:(7)$$\begin{array}{l} \varphi (r,\omega )=\varphi (r,\omega_{0})+\frac{\partial \varphi (r,\omega )}{\partial \omega }{|}_{\omega =\omega_{0}}(\omega -\omega_{0})\\ \;\;\;+\frac{\partial^{2}\varphi (r,\omega )}{2!\partial \omega^{2}}{|}_{\omega =\omega_{0}}{\left(\omega -\omega_{0}\right)}^{2}+\ldots \\ \;\;\;+\frac{\partial^{n}\varphi (r,\omega )}{n!\partial \omega^{n}}{|}_{\omega =\omega_{0}}{\left(\omega -\omega_{0}\right)}^{n}+R_{n}(\omega ) \end{array},$$
where $R_{n}\left(\omega \right)$ the *n*-th order Taylor remainder. Equation (7) shows that, for the metalens to achieve achromatic focusing near the designed frequency *ω*_0_, the nanoantenna unit cell placed at the radial coordinate r not only needs to provide the incident light the additional phase $\varphi \left(r,\omega_{0}\right)$ required, but also needs to satisfy the subsequent partial derivative terms. Among them, the first-order partial derivative of the phase distribution $\varphi \left(r,\omega \right)$ related to the angular frequency *ω* is called group delay (GD), and the second-order partial derivative is called group delay dispersion (GDD) [19]. GD and GDD play a decisive role in the dispersion compensation of a metalens. If only the phase distribution at the designed frequency is considered, and GD and GDD are ignored, it leads to chromatic aberration effects [19]. For our scheme, Equation (6) can be rewritten as
(8)$$\varphi \left(r,\omega \right)=\frac{\omega }{c}\left\{n_{0}\left[H-h_{l}\left(r\right)\right]+n_{1}h_{l}\left(r\right)\right\},$$
where *c* is the speed of light. The corresponding GD and GDD can be calculated as
(9)$$\left\{\begin{array}{l} \frac{\partial \varphi \left(r,\omega \right)}{\partial \omega }=\frac{1}{c}\left\{n_{0}\;\left[H-h_{l}\left(r\right)\right]+n_{1}\;h_{l}\left(r\right)\right\}\\ \frac{\partial^{2}\varphi \left(r,\omega \right)}{\partial \omega^{2}}=0 \end{array}\right..$$

Obviously, the GD that should be satisfied is a function only related to the radial coordinate *r*, and the GDD does not affect the dispersion compensation of our scheme. Therefore, to achieve achromatic focusing of the incident light, the array of nanorods arranged on the substrate needs to satisfy two constraints. One is that the phase distribution satisfies Equation (8), and the other is that the provided GD satisfies Equation (9).

The schematic diagram of our designed wide-angle and wide-band achromatic lens is shown in Figure 2a. After a beam of mid-infrared broadband light is incident on the metalens, the emergent light components with different frequencies can be focused at the same point on the focal plane. The incident light is linearly polarized in the *x*-direction, the center frequency is 70.54 THz (4.25 μm), and the bandwidth is 21 THz (3.7–5 μm). The designed focal length of the metalens is 200 μm. When the incident angle *θ* of the beam varies within ±30°, the emergent beam can always be efficiently focused on the focal plane. The focus is shifted to varying degrees along the *x*-axis, while the light components of each frequency remain focused on the same point.

In the design of metasurfaces involving dispersion compensation, various types of dielectric nanorods are used as the unit cell antennas. The dielectric nanorods can be considered as truncated waveguides, and the additional phase that a single antenna provides can be expressed as follows [19]:(10)$$\varphi \left(r,\omega \right)=\frac{\omega }{c}n_{eff}g,$$
where *n_eff_* is the effective refractive index of the waveguide, and *g* is the height of the nanorod. Therefore, the GD possessed by a dielectric nanorod can be calculated as follows [18]:(11)$$\frac{\partial \varphi \left(r,\omega \right)}{\partial \omega }=\frac{1}{c}n_{eff}g+\frac{\omega }{c}\frac{\partial n_{eff}}{\partial \omega }g=\frac{g}{\frac{\partial \omega }{\partial k}}=\frac{g}{v},$$
where *k* is the waveguide constant, and *v* is the group velocity. By changing the size and duty cycle of the dielectric nanorod to control the waveguide constant *k*, or adjusting the height *g* of the nanorod, the GD possessed by the nanoantenna can be freely regulated. In previous reports, Ou et al. conducted a comprehensive and systematic study of dielectric nanorod materials, as well as antenna shapes and sizes, suitable for mid-infrared wavelengths [16,23]. The monocrystalline silicon nanorod structure has the advantages of high transmittance, low loss, obvious waveguide effect, and large phase adjustment amplitude in the mid-infrared band; thus, it is an excellent choice for antennas of metasurface in this band. In addition, the elliptical nanorod is adopted, and the lengths of the major and the minor axes of the ellipse serve as two adjustable degrees of freedom, ensuring a large range of GD variation. The construction of our nanorods database is based on part of their work, and we also conducted a more in-depth study.

The arrangement of the nanorod arrays on the metalens and the structural parameters of the nanorods can be clearly observed from Figure 2b. In order to arrange as many unit cells as possible, the elliptical nanorods are arranged in polar coordinates on the substrate, forming a metalens. The radius of the metalens *R* is 50.5 μm, and the numerical aperture is 0.245. The enlarged image on the right shows the parameters of a single unit cell. Both the substrate and the nanorods are made of monocrystalline silicon. The refractive index of monocrystalline silicon in the working bandwidth changes from 3.427 to 3.422 [38]. The nanorod is placed in the center of a square lattice, and the lattice period is selected as *p* = 1650 nm. The height of the nanorod *G* is fixed to 4.5 μm; the long axis of the elliptical section is denoted by *r_x_*, and the short axis is denoted by *r_y_*. We use the commercial software Finite Difference Time Domain (FDTD) Solutions to scan the geometric parameters of the nanorods by simulation. During the scanning process, both the long axis *r_x_* and the short axis *r_y_* of the ellipse change from 200 nm to 700 nm with 251 uniform sampling points. The incident light is an *x*-polarized mid-infrared broadband (3.7–5 μm) plane wave. Periodic boundary conditions are set in both *x*- and *y*-directions, while the perfect matching layer (PML) is set in the *z*-direction (the incident direction of the beam). From the scanning results, a complete database of transmittance and phase spectra of monocrystalline silicon elliptical nanorods with different sizes in the working spectrum is established. We present the *r_x_*- and *r_y_*-dependent phase and transmittance datasets at the center frequency (corresponding to a wavelength of 4.25 μm) in Figure 2c. This part of the data is particularly important and is directly used in the subsequent structure search procedure. As can be seen from the figure, in the adopted database, a phase change of 2π is sufficient. The transmittance of each structure is greater than 0.6, ensuring high intensity of the emergent light. By fitting the phase *φ* to a linear function of the angular frequency *ω*, the GD of nanorods with different sizes was obtained, as shown in the lower left panel of Figure 2c. The relative GD of all structures varied from 0 to 49.88 fs. We also performed a linearity fit test (*R*-squared) for each GD value as in [19]. The high *R*-squared value indicates that the structural dispersion of the nanorods was significant, which is beneficial for the metalens to achieve continuous achromatic behavior in a wide waveband. The test results are shown in the lower right panel of Figure 2c. In the working bandwidth, the *R*-squared values corresponding to all GD values were greater than 0.99.

### 2.3. Design Method for Determining the Structure of Metalens

Our metalens has M = 31 circles of nanorods arranged from the center to the edge of the metalens. All the nanorods in the same circle have the same parameters, while the parameters of the nanorods in different circles are different. We used a search procedure to find elliptical nanorods that satisfy both the required phase and the GD values at each pixel of the metalens; then, their major and minor axis lengths were determined. The search procedure was adapted and refined on the basis of the search algorithm devised by our previous work [39]. We wrote this algorithm for finding nanorods using MATLAB software. The complete flowchart of the search procedure is shown in Figure 3a. In the search algorithm, we comprehensively consider the phase, GD, and transmittance of the nanorods. Therefore, we need three datasets to provide these simulation data, namely, the phase spectrum at the center frequency, the transmittance spectrum at the center frequency, and the GD spectrum in Figure 2c. In the flowchart, they are denoted as φ, t, and GD, respectively. The number of elliptical nanorods with different sizes included in the dataset is L × S, in which L = 251 corresponds to the long axis rx, and S = 251 corresponds to the short axis ry. In addition, we also need to calculate the phase φreq (m) and group delay GDreq (m) that the unit cell needs to provide for each circle on the metalens. Among them, m ∈ [1, 31] represents the unit cell of the m-th circle from the center to the edge of the metalens. φreq (m) and GDreq (m) are calculated from Equations (8) and (9), respectively, where r = (m − 1) × P, i.e., the radial coordinate corresponding to the center of the lattice of the m-th unit cell. The main principle of the search algorithm is to minimize the difference between the required phase (φreq (m))/group delay (GDreq (m)) and the phase (φact (m))/group delay (GDact (m)) provided by the actual selected nanorods.

Since *φ* and GD in the original datasets of Figure 2c are relative phases and GD, it is not convenient to use these data in the search program directly, we need to first choose the most suitable structure in the datasets and set it as the origin. The phase and GD of the origin point are adjusted to the phase and GD required by the unit cell in the first circle (at the center) of the metalens, i.e., *φ_req_* (1) and *GD_req_* (1). Other structures in the datasets are also adjusted by the same amount simultaneously. The adjusted datasets, which include *φ_sim_* and *GD_sim_*, are used as input of the search algorithm. This part of the calculation can be expressed as
(12)$$\begin{array}{c} \varphi_{sinm}=\varphi -\varphi \left(l,s\right)+\varphi_{req}\left(1\right)x\\ GD_{sim}=GD-GD\left(l,s\right)+GD_{req}\left(1\right) \end{array},$$
where *l*, *s* ∈ [1, 251], which means that we sequentially set each structure in the datasets as the origin point.

Through the first step, the structure set as the origin point is naturally selected as the unit cell of the first circle of the metalens, since its phase and GD are tuned to fully meet the design requirements. However, starting from the second unit cell, the search is no longer easy, as there may not be a structure in the entire datasets that fully satisfies the desired phase and GD simultaneously. For the *m*-th unit cell, the error between the required phase and the phase provided by each structure in the datasets, as well as that corresponding to the GD, can be expressed as
(13)$$\begin{array}{c} D_{\varphi }\left(m\right)=\left|\varphi_{req}\left(m\right)-\varphi_{sim}\right|\\ D_{GD}\left(m\right)=\left|GD_{req}\left(m\right)-GD_{sim}\right| \end{array},$$
where *m* ∈ [1, 31], and *D_φ_* (*m*) and *D_GD_* (*m*) are the *L* × *S* (251 × 251) error matrices. We use these error matrices to find structures with sufficiently small errors to ensure that the constructed metalenses perform their designed functions.

In order to solve the above problem, we defined a series of error levels to limit the tolerable errors, as shown in Table 2. Each level is composed of a phase error factor *B_φ, i_* and a GD error factor *B_GD, i_*, which means that the maximum tolerable error for this level is *B_φ, i_* (*B_GD, i_*) × *φ_err_* (*GD_err_*). *φ_err_* and *GD_err_* respectively represent the maximum values in all error matrices *D_φ_* (*m*) and *D_GD_* (*m*) calculated by Equation (13). The computation of these two values includes each structure in the datasets as the origin point. A higher level means a larger tolerable error. After the search cycle starts, the error tolerance level is first set to 1, i.e., the tolerable error is the smallest. Then, the error matrices *D_φ_* (*m*) and *D_GD_* (*m*) are traversed to find all structures that satisfy both *D_φ_* (*m*) ≤ *B_φ,_*
_1_ × *φ_err_* and *D_GD_* (*m*) ≤ *B_GD,_*
_1_ × *GD_err_*, which are marked as [*u*, *v*]. As long as the number of structures found in this traversal is not 0, the search ends. When more than one eligible structure is found in a traversal, the program compares the transmittances by looking up the transmittance dataset, and leaves the highest one. If this traversal does not find a suitable structure, the error tolerance level is increased to 2, 3, 4, … and the matrix is re-traversed until a structure that satisfies the condition is found. A total of nine different levels are set in our program. It can be seen that *B_φ, i_* and *B_GD, i_* are not equally weighted in the latter grades; GD requires greater error tolerance. This shows that, for our metalens, the GD at each position is more difficult to satisfy perfectly. As can be seen later, our final search results also reflect this feature. The setting of the weight ratio of *B_φ, i_* and *B_GD, i_* requires repeated deliberation and combination of subsequent device simulation results. We tried many different weighting ratios. Using the error factors shown in Table 2, the best overall device performance of all attempts was obtained.

Through a single-cycle search of *m* ∈ [1, 31], we can determine the corresponding structure of each unit cell. The transmittances of each selected structure are then summed to obtain *T* (*l*, *s*). However, this set of structures is only the search result with the structure (*l*, *s*) as the origin point. We need another double cycle search of *l*, *s* ∈ [1, 251] so that each structure acts as the origin point. Finally, 251 × 251 *T* values can be obtained, which correspond to 251 × 251 sets of structures that are initially screened. Screening is repeated by comparing the magnitude of these *T* values and select the largest one *T* (*l*_0_, *s*_0_). In this way, the set of structures filtered out when the origin point is (*l*_0_, *s*_0_) is the final result of the search procedure. The geometrical dimensions (*r_x_* and *r_y_*) of each unit cell are determined according to the results of the search algorithm, and then the metalens simulation model shown in Figure 2b is established using Lumerical FDTD Solutions 2018a. Figure 3b shows the error margin for the 31 nanorods found by the search algorithm. The error margin in phase and GD of most nanorods were less than 0.5% of the theoretical maximum. The GD error of a small number of nanorods was larger, but all cases were less than 5% of the theoretical maximum. This shows that the search algorithm could accurately find the nanorods in the database with the desired phase and GD.

Figure 4a shows the required phase and GD (*φ_req_* (*m*) and *GD_req_* (*m*)) profiles of the achromatic metalens along the *x*-axis, compared with the phase, as well as the GD actually provided by the nanorods (*φ_act_* (*m*) and *GD_act_* (*m*)) found by the search algorithm. In order to better demonstrate the performance of our designed wide-field and wide-band achromatic focusing metalens, a conventional metalens is also designed as a contrast. This metalens has the same device parameters as the achromatic lens, but the lens follows the phase distribution expressed by Equation (1). Thus, although this metalens has no spherical aberration, there is still a large number of monochromatic aberrations. On the other hand, the algorithm flow shown in Figure 3a is also used to find the cell structure of this lens, the difference is that no constraint is imposed on GD in the algorithm. Therefore, a metalens with chromatic aberration is finally designed. The required phase and GD profiles of the chromatic lens along the *x*-axis and the actual phase and GD of the nanorods found by the algorithm are shown in Figure 4b. In the two figures, the red curve is the theoretical phase profile, and the blue curve is the theoretical GD profile. From the two figures, we can see that, due to the tenth-order aspheric structure used in the aberration correction step, the phase and GD profiles of the wide-angle achromatic lens are no longer the same quadratic curve form as the chromatic lens. The magnitude of change in phase and GD is also different from that of the chromatic lens. The red and blue dots in the two figures are the actual phase and GD provided by the nanorods found by the search algorithm, respectively. Due to the constraints of the phase error factor, the actual phase distribution of the two lenses almost coincides with the theory. The actual GD profile of achromatic lens deviates slightly from theory. Under the constraint of the GD error factor, the shape of the profile is basically consistent with the theory, and the error is within the tolerable range. In contrast, the chromatic lens has no GD constraint, resulting in the actual GD of most nanorods being far from the theoretical curve. Lastly, we show in Table 3 the metalens parameters and structural dimensions (*r_x_* and *r_y_*) of each circle of nanorods from the center to the edge of the achromatic lens and the chromatic lens determined by the search algorithm, as well as their distances *R_m_* from the center of the metalens (*m* is the number of circles).

## 3. Simulation Results and Discussion

We then calculated the focusing performance of the achromatic and the chromatic lenses using FDTD solutions. We first simulated the case where the mid-infrared broadband light (3.7–5 μm) was incident parallel to the optical axis. In order to visually analyze the focusing effect of the metalens, we used the far-field script function based on Equation (2) in FDTD Solutions to obtain the electric field intensity distributions in the *x*–*z* plane for different light wavelengths of the achromatic and chromatic lenses. The results are shown in Figure 5a,c, respectively. Each intensity distribution map is normalized by the maximum intensity in the respective map to show the actual focus position of the emergent light more clearly under different light wavelengths. It can be seen from Figure 5a that the focal points of the emergent beams at different wavelengths were almost at the same focal length. The maximum intensity position of the focal point at each wavelength fluctuated up and down to a certain extent, but all cases were within the focal tolerance of the lens. The lens exhibited remarkable achromatic focusing properties. It is a little regrettable that the actual focal length of the lens at each wavelength was slightly larger than the designed focal length (200 μm), marked with a white dashed line in the figure. In contrast, Figure 5c exhibits the chromatic focusing characteristics. As the wavelength of the incident light increased, the actual focal length of the lens continued getting longer. At short wavelengths, the actual focal length of the lens was less than 200 μm, while, at long wavelengths, it was greater than 200 μm. Moreover, as the wavelength increased, the depth of focus also increased significantly.

We also simulated the *x*–*y* plane field distributions on the designed focal plane (at 200 μm) to better describe the focused beam, and the results are shown in Figure 5b,d. In order to show the intensity difference of the focus more intuitively under different light wavelengths, each intensity distribution map was normalized by the maximum intensity of the full working band. The focal spots of both the achromatic and the chromatic lenses at different wavelengths show circularly symmetric intensity distributions. We cannot conclude that the focusing power of the achromatic lens on the designed focal plane was superior to that of the chromatic lens when the incident angle *θ* of the beam was 0°. In fact, the chromatic lens was more evenly focused overall on the working band. The achromatic lens was more diffraction-limited in the focus at short wavelengths but significantly defocused at the long wavelengths. The error between the was and the designed focal length of the achromatic lens in this part of the waveband is larger than that of the chromatic lens, which led to this phenomenon. We believe that this problem was mainly due to in the monochromatic aberration correction step; in order to balance the overall performance of the lens in a wide FOV, especially to significantly reduce the focal aberration of the lens when the incidence angle *θ* is close to 30°, the optimized merit function in Zemax sacrifices some of the focusing quality of the lens near the paraxial axis. While this effect is not perfect, we consider it reasonable and acceptable. The focusing efficiency of the metalens was more balanced in the entire FOV, showing a clear achromatic advantage at a larger FOV.

To verify this analysis, we performed extensive simulations of the focusing effect of the achromatic and the chromatic lenses at beam incidence angles other than 0° (*θ* continuously increasing from 0° to 30°). In Figure 6, we show representative far-field intensity profiles for *θ* = 30°. Simulation results from other angles are later presented in the form of data plots. Figure 6a,c show the *x*–*z* plane light intensity distributions of the achromatic and the chromatic lenses. We can find that the focal spot was inclined at each wavelength and had different degrees of offset relative to the optical axis along the *x*-direction. The actual focus of the achromatic lens at each wavelength was very close to the designed focal length of 200 μm, and the focal depth was relatively consistent, showing good achromatic performance. However, the focal length and the focal depth of the chromatic lens still became larger with the increase in wavelength. Moreover, the actual focal length was significantly smaller than the designed focal length in most wavebands.

Figure 6b,d depict the *x*–*y* field distributions of the achromatic and the chromatic lenses at the designed focal plane. In order to visually show the difference between the focal intensity of the metalens at a large FOV and the paraxial incidence, each distribution map was normalized by the maximum focal plane intensity in the full working band when the beam incident angle *θ* was 0°. We can see that, even at the center frequency, the maximum intensity at the focal plane was less than 1, which reflected a larger focal aberration at oblique incidence than that at perpendicular incidence. Nonetheless, the monochromatic aberration-corrected metalens maintained a comparable focal intensity near the center frequency, while the uncorrected metalens had very weak focal intensity. The simulation results clearly show that the optical design optimized the phase distribution; thus, the aberration of the metalens was significantly reduced under the condition of large FOVs, and the metalens could work in a wide FOV up to 60°. In contrast, traditional metalenses are difficult to focus due to the excessive aberrations at large FOVs, which struggle to meet practical needs. In fact, we can find from the simulation results that, at large FOVs, the focal intensity of the achromatic lens was significantly larger than that of the chromatic lens in the entire working waveband, although the focal intensity of the achromatic lens decreased at the edge band. We believe that this was due to the combined action of monochromatic aberration correction and dispersion compensation.

To better understand the balance of focusing effect of the designed metalens in a wide FOV and a wide spectral range, we sorted out the actual focal length of the achromatic and the chromatic lenses at different FOVs and different wavelengths from the simulation results. We also calculated performance evaluation indicators such as the full width at half maximum (FWHM) of the focal spot, the focusing efficiency, and the Strehl ratio of the lens. It is worth noting that, in order to maintain consistency, the calculation of these evaluation indicators is all aimed at the focal spot on the designed focal plane (200 μm), rather than the actual focal point of the metalens at each wavelength and each FOV. The results are shown in Figure 7. Figure 7a shows the actual focal lengths of the achromatic and the chromatic lenses at different angles of inclination *θ* and different wavelengths of the incident light. The solid and dashed lines in the figure represent the simulation results for the achromatic and the chromatic lens, respectively. The red, green, blue, and cyan lines represent the inclination angle corresponding to *θ* = 0°, 10°, 20°, and 30°, respectively. The designed focal length of the metalens (200 μm) is also indicated by the black dashed line in the figure. We can find that, when the incident angle *θ* was 0° and 10°, the actual focal length of the chromatic lens on average was closer to the designed focal length than the achromatic lens over the entire waveband. However, as the incident angle *θ* increased to 20° and 30°, the actual focal length of the achromatic lens was overall closer to the designed focal length. Among the eight groups of results, the results obtained by the achromatic lens at the incident angle of 30° were the closest to the designed focal length. The simulation results also show that, by using the method of optical design to correct the aberration, the focus quality of the metalens at large angles of incidence, especially in the fringe FOV, was improved. Correspondingly, the focusing effect in the center of the FOV decreased to a certain extent.

Figure 7b shows the FWHM of the focal spots for both the achromatic and the chromatic lenses. In addition, the ideal diffraction limit of the metalens is calculated as follows [40]:(14)$$d=\frac{\lambda f}{D},$$
where *D* is the diameter of the metalens. The calculated results are represented by the black dashed lines in Figure 7b. The results show that the chromatic lens performed well at smaller FOVs (0° and 10°), with the FWHM of the focal spots very close to the diffraction limit. However, with the increase in the FOV, the FWHM increased significantly; especially in the shorter waveband, the FWHM value of multiple sampling points was several times higher than the ideal diffraction limit, reflecting very large aberrations. For the achromatic lens, despite the correction of monochromatic aberrations, the FWHM of the focal spots was still larger than the corresponding theoretical limits at different incident angles. With the increase in the incident angle, the FWHM of the focal spots at different wavelengths also increased to a certain extent, indicating that the monochromatic aberration of the metalens was not completely eliminated. We believe that this is because our design method considers the correction of both monochromatic aberration and chromatic aberration; thus, the final optimization result cannot perfectly eliminate a certain aberration but pursues the overall reduction in various aberrations to achieve the best result. Therefore, we can find that the FWHM difference of the achromatic lens at different incident angles and different wavelengths was smaller than that of the chromatic lens. When the FOV reached 30°, the FWHM values were significantly better than the chromatic lens. The results show that, by combining the monochromatic aberration correction scheme in [37] with the dispersion compensation method in [19], the balance of the imaging performance of the metalens could be improved, and the difference in the output of the metalens in different FOV and different wavebands could be reduced, thereby improving the overall performance. In particular, the metalens optimized by our method can maintain good performance for larger incident angles.

In addition, we also calculated the focusing efficiencies [31]. The calculated results are shown in Figure 7c. The results show that the chromatic lens had a high focusing efficiency near the center frequency when the incident angle *θ* was small. However, at the edge band, the focusing efficiency of the lens dropped significantly. The lens cannot maintain a relatively constant focusing efficiency over a wide spectral range. More seriously, as *θ* increased, including the center frequency, the focusing efficiency of the metalens dropped fast to a low value at all wavelengths. On the contrary, the focusing efficiency of the achromatic lens did not change much under the conditions of different wavelengths and inclination angles. At *θ* = 30°, the focusing efficiency of the achromatic lens was higher than that of the chromatic lens over the entire working band. The focusing efficiency of the achromatic lens was 6.7–42.4% in the working bandwidth of 3.7–5 μm and the FOV of 60°. This is comparable to the 12.44–68% focusing efficiency of previously reported achromatic metalenses [18,19,21,22,23,24,31,41], and it outperforms some previously reported monochromatic aberration-corrected metalenses [30,37]. It can be seen that the advantage of our method is that the designed metalens has an overall balance of focusing power and can be applied in more environments than a single achromatic or aberration-optimized metalens.

Lastly, we also calculated the Strehl ratios of the focal spots, and the results are shown in Figure 7d. The Strehl ratio is defined as the peak intensity of the point spread function (PSF) of the focal spot at the designed focal plane [42]. The PSFs of the focal spots of the metalens at different incident light wavelengths and incident angles are normalized by the PSF peaks of the aberration-free lens with the same power. A lens is considered to be diffraction-limited when the Strehl ratio is greater than 0.8. The experimental results are similar to the previous evaluation metrics. The Strehl ratio of the chromatic lens is high when the beam incident angle *θ* is small. Even in some wavebands, the lens can be considered to be diffraction limited when the Strehl ratio is greater than 0.8. The experimental results are similar to the previous evaluation metrics. The Strehl ratio of the chromatic lens was high when the beam incident angle *θ* was small. Even in some wavebands, the lens can be considered to be diffraction-limited. However, with the increase in the incident angle, the value of Strehl ratio decreased obviously. When *θ* was 30°, the Strehl ratio value of the chromatic lens was inferior to that of the achromatic lens in the entire working band, showing a sharp deterioration of its aberration at the fringe FOV. The Strehl ratio of the achromatic lens in each FOV and each wavelength band was relatively stable, remaining between 0.314 and 0.75. The results again confirm that the metalens with a balanced focusing effect and high efficiency in a large FOV and a wide working band can be designed using a convenient optical design combined with dispersion engineering and a direct search algorithm [19,37], rather than a machine learning process or the help of high-performance computing services.

## 4. Conclusions

In conclusion, a general design method for single-layered metalenses with both achromatic and wide-angle focusing capability was proposed. In the design flow, we first obtained the monochromatic aberration-corrected phase profile using the optical design, and then conducted a direct search algorithm to find elliptical nanoantennas with a suitable phase and GD directly in the material library. Using this method, we designed a metalens with balanced and high focusing efficiency in the mid-infrared band (3.7–5 μm) and a wide FOV (60°). The highest focusing efficiency of the metalens in the entire working band and FOV was 42.4%, and the average efficiency was 21.84%. In addition, it was demonstrated that the focusing effect of our metalens was significantly better than that of the conventional metalens at a large incident angle of 30°. The designed metalens has a simple structure and low fabrication difficulty and is expected to be applied in compact and miniaturized mid-infrared detection and imaging systems. The proposed method of designing metalenses has the advantages of a simple design principle, low computational load, and wide application range, paving the way for the development of programmatic design and optimization software for metasurfaces in the future.

## Figures and Tables

**Figure 1 materials-15-07587-f001:**
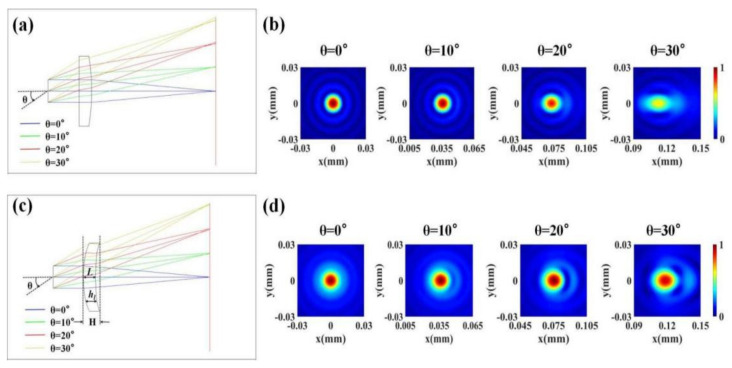
(**a**,**c**) The focusing effect of the lens (**a**) without and (**c**) with monochromatic aberration corrected at different incident angles. The blue, green, red, and yellow lines represent the propagation trajectories of rays with incident angles θ = 0°, 10°, 20°, and 30°, respectively. L in (**c**) is the thickness at the center of the lens, hl represents the actual thickness at different positions of the lens, and the total thickness of the system is denoted by H. (**b**,**d**) The electric field intensity distributions at the focal plane of the lens (**b**) without and (**d**) with monochromatic aberration-corrected at different incident angles. All electric field intensity profiles are normalized. The focal points are all placed in the central area of the images, and the actual x-direction offset is used to distinguish different incident angles.

**Figure 2 materials-15-07587-f002:**
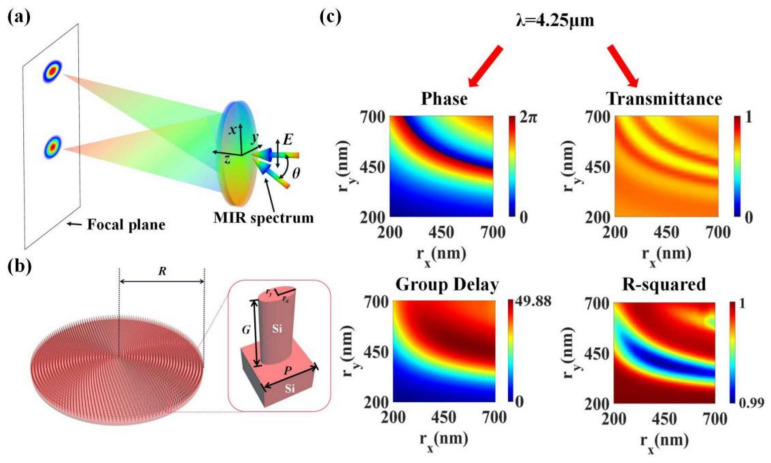
(**a**) Schematic illustration of a wide-FOV and broadband achromatic focusing metalens. The mid-infrared broadband (3.7–5 μm) *x*-polarized light is incident on the metalens from the substrate. The transmitted light is focused achromatically on the focal plane. When the inclination angle *θ* of the incident light varies within ±30°, the aberration-corrected metalens can still focus the beam on this focal plane. (**b**) Top view of the metalens and a magnified view of a unit cell on the metalens. The radius of the metalens is *R* = 50.5 μm. The unit cell is made of all monocrystalline silicon. The lattice period is set as *p* = 1650 nm. The height of the elliptical nanorod is *G* = 4.5 μm, and the long axis *r_x_* and the short axis *r_y_* are two variable structural parameters. (**c**) Top left image: Phase spectrum at the center frequency (corresponding wavelength is 4.25 μm) in the nanorods database. Top right image: Transmittance spectrum at the center frequency in the nanorods database. Bottom left image: GD values of nanorods with different sizes obtained by fitting. Bottom right image: *R*-squared value corresponding to each GD value.

**Figure 3 materials-15-07587-f003:**
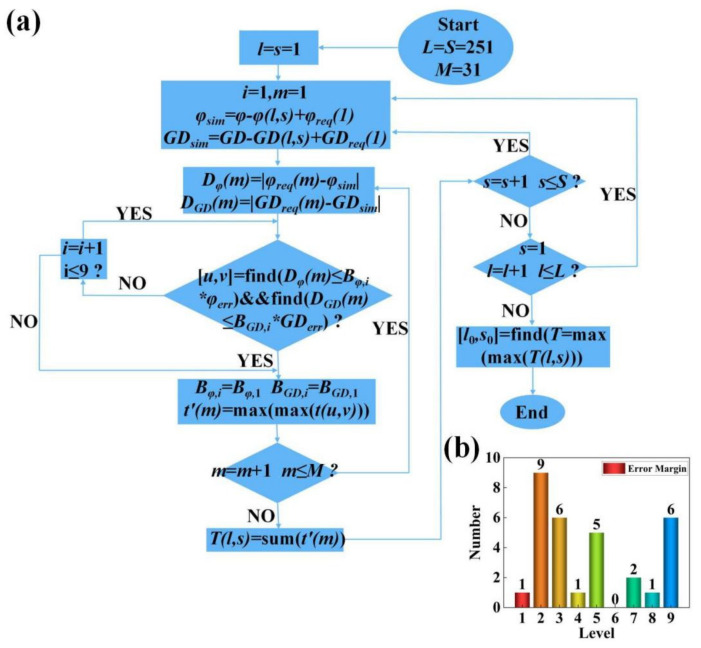
(**a**) Algorithmic flowchart of the nanorod search procedure. (**b**) Error margin of the 31 nanorods selected using the search procedure.

**Figure 4 materials-15-07587-f004:**
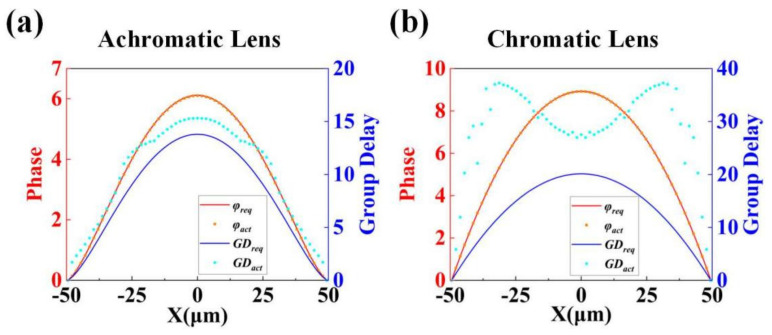
Theoretical phase and GD curves along the *x*-axis and the actual phase and GD values of the selected nanorods for (**a**) the achromatic lens and (**b**) the chromatic lens. Red curve: theoretical phase values; blue curve: theoretical GD values; red dots: actual phase values; blue dots: actual GD values.

**Figure 5 materials-15-07587-f005:**
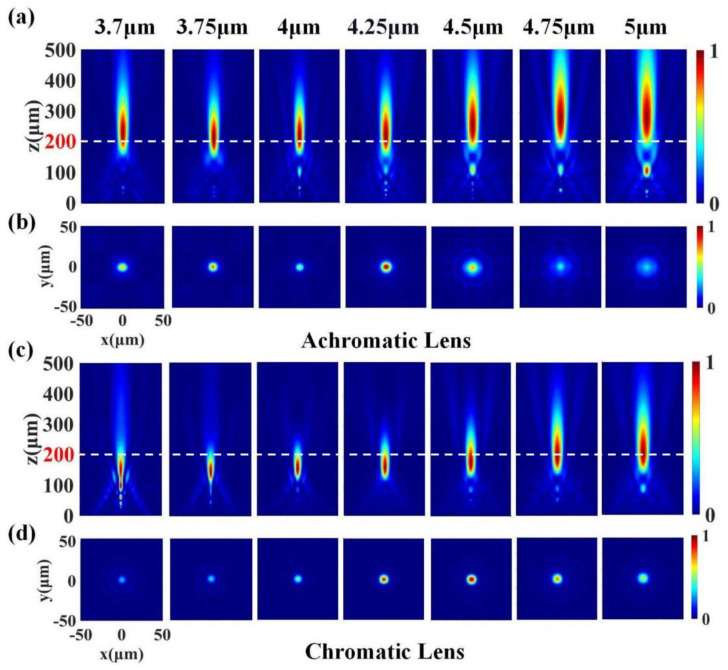
Simulation results within the working bandwidth of 3.7–5 μm. The angle *θ* between the incident light and the optical axis is 0°. (**a**) Normalized *x*–*z* plane light intensity distributions for seven different wavelengths of the achromatic lens. (**b**) Normalized *x*–*y* plane light intensity distributions on the designed focal plane (at 200 μm), marked with a white dashed line in (**a**). (**c**) Normalized *x*–*z* plane light intensity distributions for seven different wavelengths of the chromatic lens. (**d**) Normalized *x*–*y* plane light intensity distributions on the designed focal plane (at 200 μm), marked with a white dashed line in (**c**). All focal points fall on the optical axis.

**Figure 6 materials-15-07587-f006:**
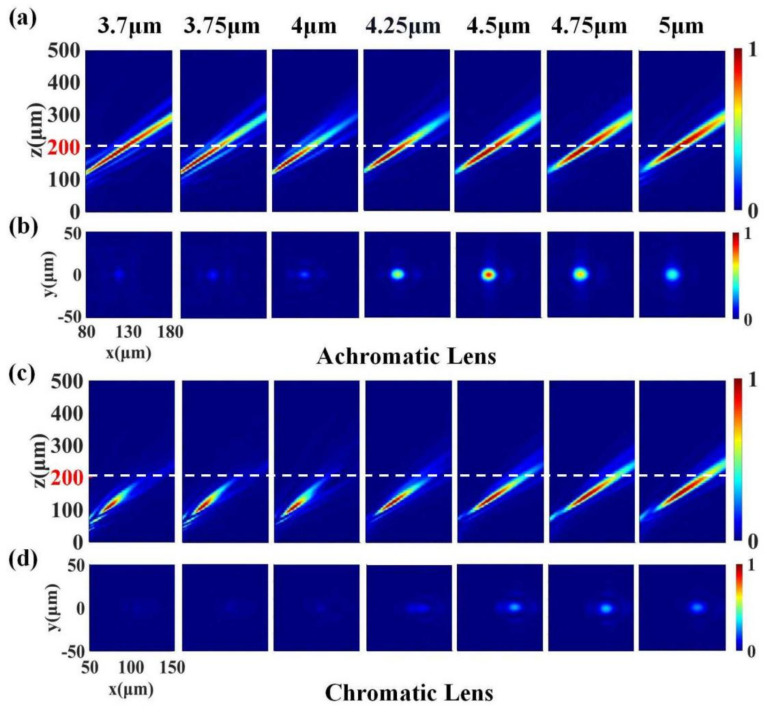
Simulation results within the working bandwidth of 3.7–5 μm. The angle *θ* between the incident light and the optical axis is 30°. (**a**) Normalized *x*–*z* plane light intensity distributions for seven different wavelengths of the achromatic lens. (**b**) Normalized *x*–*y* plane light intensity distributions on the designed focal plane (at 200 μm), marked with a white dashed line in (**a**). (**c**) Normalized *x*–*z* plane light intensity distributions for seven different wavelengths of the chromatic lens. (**d**) Normalized *x*–*y* plane light intensity distributions on the designed focal plane (at 200 μm), marked with a white dashed line in (**c**). All focal points are offset from the optical axis in the *x*-direction to varying degrees.

**Figure 7 materials-15-07587-f007:**
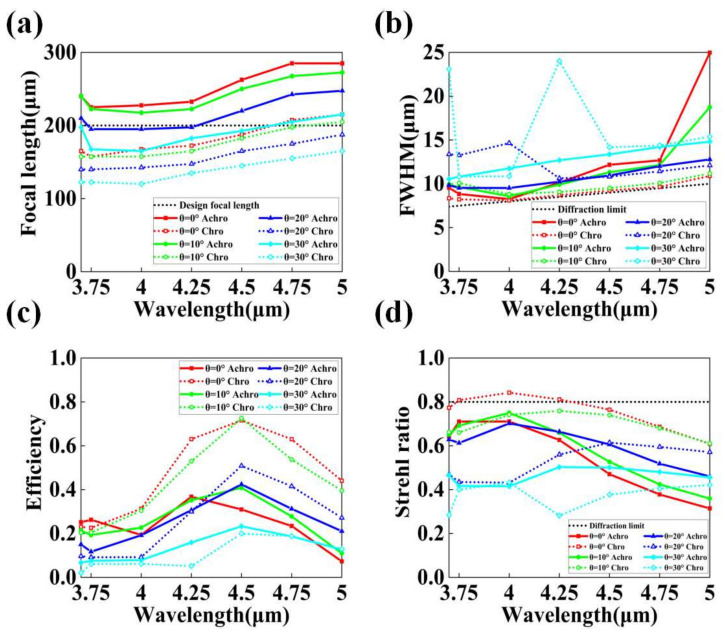
(**a**) Actual focal lengths of the achromatic and the chromatic lenses at different incident angles *θ* and different wavelengths. The black line represents the designed focal length (200 μm) of the metalens. (**b**) The FWHM of the focal spots for the achromatic and the chromatic lenses at different incident angles *θ* and wavelengths. The black line represents the ideal diffraction limit of the metalens. (**c**) Focusing efficiencies of the achromatic and the chromatic lenses at different incident angles *θ* and different wavelengths. (**d**) Strehl ratios of the focal spots for the achromatic and the chromatic lenses at different incident angles *θ* and different wavelengths. The black line represents the minimum value of the Strehl ratio (0.8) that can be considered a diffraction limited lens. In all figures, solid lines represent simulation results for the achromatic lens, and dashed lines represent simulation results for the chromatic lens. The red, green, blue, and cyan lines represent the inclination angle *θ* = 0°, 10°, 20°, and 30° of the incident light, respectively.

**Table 1 materials-15-07587-t001:** Parameters of the aspherical lens after monochromatic aberration optimization using Zemax.

	1	2
$c\left(mm^{-1}\right)$	1/0.199707	1/0.620348
K	−0.706966	115.017417
$a_{1}$	0	0
$a_{2}$	393.991442	471.989551
$a_{3}$	1.870081 × 10^4^	2.363506 × 10^4^
$a_{4}$	3.710574 × 10^6^	5.501649 × 10^6^
$a_{5}$	3.882496 × 10^8^	3.882496 × 10^8^

**Table 2 materials-15-07587-t002:** Error factor values for each error level.

i	1	2	3	4	5	6	7	8	9
*B_φ, i_*	0.0001	0.001	0.002	0.003	0.005	0.005	0.005	0.005	0.005
*B_GD, i_*	0.0001	0.001	0.002	0.003	0.005	0.01	0.02	0.03	0.05

**Table 3 materials-15-07587-t003:** The metalens parameters of unit cells for the achromatic lens and the chromatic lens.

The Metalens Parameters											
Incident light wavelength (λ)								3.7–5 μm			
Radius of monocrystalline silicon substrate (*R*)								50.5 μm			
Monocrystalline silicon nanopillar height (*H*)								4.5 μm			
Long axis radius (*r_x_*) and short axis radius (*r_y_*) of monocrystalline silicon nanopillar								200 nm *≤ r_y_,r_y_ ≤* 700 nm			
Design focal length (*L*)								200 μm			
Lattice period (*P*)								1650 nm			
**Achromatic Lens**											
*m*	*r_x_* (nm)	*r_y_* (nm)	*R_m_* (μm)	*m*	*r_x_* (nm)	*r_y_* (nm)	*R_m_* (μm)	*m*	*r_x_* (nm)	*r_y_* (nm)	*R_m_* (μm)
**1**	532	648	0	**11**	696	330	16.50	**21**	682	232	33.00
**2**	526	666	1.65	**12**	698	320	18.15	**22**	682	686	34.65
**3**	520	680	3.30	**13**	700	310	19.80	**23**	642	230	36.30
**4**	574	532	4.95	**14**	698	302	21.45	**24**	618	232	37.95
**5**	592	492	6.60	**15**	700	292	23.10	**25**	592	238	39.60
**6**	612	454	8.25	**16**	700	282	24.75	**26**	568	244	41.25
**7**	634	418	9.90	**17**	698	272	26.40	**27**	536	258	42.90
**8**	652	390	11.55	**18**	696	262	28.05	**28**	504	278	44.55
**9**	670	366	13.20	**19**	700	248	29.70	**29**	368	582	46.20
**10**	682	348	14.85	**20**	698	236	31.35	**30**	400	438	47.85
								**31**	400	420	49.50
**Chromatic Lens**											
*m*	*r_x_* (nm)	*r_y_* (nm)	*R_m_* (μm)	*m*	*r_x_* (nm)	*r_y_* (nm)	*R_m_* (μm)	*m*	*r_x_* (nm)	*r_y_* (nm)	*R_m_* (μm)
**1**	620	648	0	**11**	662	460	16.50	**21**	516	454	33.00
**2**	602	690	1.65	**12**	618	498	18.15	**22**	626	294	34.65
**3**	650	582	3.30	**13**	534	654	19.80	**23**	478	468	36.30
**4**	604	672	4.95	**14**	526	642	21.45	**24**	600	272	37.95
**5**	606	656	6.60	**15**	502	682	23.10	**25**	680	212	39.60
**6**	600	656	8.25	**16**	520	586	24.75	**26**	388	640	41.25
**7**	622	592	9.90	**17**	490	652	26.40	**27**	432	426	42.90
**8**	572	692	11.55	**18**	480	652	28.05	**28**	488	286	44.55
**9**	562	696	13.20	**19**	468	662	29.70	**29**	512	224	46.20
**10**	556	686	14.85	**20**	474	600	31.35	**30**	308	670	47.85
								**31**	374	290	49.50

## Data Availability

The data generated and/or analyzed during the current study are not publicly available for legal/ethical reasons but are available from the corresponding author on reasonable request.

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
