# Peer review of "Mid-Infrared Broadband Achromatic Metalens with Wide Field of View"

_materials, 2022, doi:10.3390/ma15217587_

Round 1

Reviewer 1 Report

The manuscript is written well with detailed design procedure. Simulation results prove that the designed metalens has large FOV and is achromatic in mid-infrared region. The introduction part could be more concise to outline the objective of this work. The authors did an excellent job in literature searching and enumerated many other works but fail to establish a connection between the drawback of previous works and the improvement of this work. 

Reviewer 2 Report

In this paper Authors propose the abberation-corrected and achromatic metalens. My prposed changes are as follows:

1. Authors  should compare the obtained simulation results with the experimental outputs.

2. Authors are missing some recent artiles in the field such as Investigation of Hyperbolic Metamaterials.

3. Authors should stress novelty of their work in comparison with others.

4. Authors should add an additional Figure describing the design under investigation.

Reviewer 3 Report

The manuscript by Jiang et al. reports on the design and numerical simulations of a broadband mid-infrared achromatic metalens. The authors use elliptical rods as the metalens unit cell of an aspheric lens and optimize the global phase and dispersion response in order to minimize chromatic aberrations over a broad wavelength range (3.7 to 5 microns) and incidence angle (0 to 60 degrees). The optimization method uses a direct search algorithm based on the individual rod response and dispersion engineering.

The results are interesting for the design of efficient and robust flat optics components, here in the mid-IR. The method is reasonably straightforward to implement, potentially applicable to other components/optimization and allow for a detailed numerical comparison with the corresponding chromatic metalens. In my opinion, this work could be published in Materials, provided the authors address the following (minor) presentation issues.

-In the introduction, the authors should precise what they mean when they introduce the "conventional metalens" for the first time (l. 125-126).
-Notation: the distance "r" between the observation point and the point of the lens surface after Eq. 2 conflicts with the radial coordinate of the lens in Eq. (1).
-The authors could explain more clearly the approximation made to get Eq. 6.
-What is the incident light polarization for the graphs of Fig. 1?
-line 463: what do the authors mean by "tolerable" range?
-line 492: the authors write that "it is a little regrettable that the actual focal lens at each wavelength is slightly larger than the designed focal length". Do they have an explanation for this? Could this systematic overshoot not be precompensated in the simulations?

Reviewer 4 Report

Please see in the pdf-file.
